# Feeding, Muscle and Packaging Effects on Lipid Oxidation and Color of Avileña Negra-Ibérica Beef

**DOI:** 10.3390/ani11102863

**Published:** 2021-09-30

**Authors:** Marta Barahona, Maria del Mar Campo, Mohammed Amine Hachemi, Maria del Mar González, José Luis Olleta

**Affiliations:** 1Department of Animal Production and Food Science, Instituto Agroalimentario (IA2), Universidad de Zaragoza-CITA, Miguel Servet 177, 50013 Zaragoza, Spain; marimar@unizar.es (M.d.M.C.); hachemi.inmv@gmail.com (M.A.H.); olleta@unizar.es (J.L.O.); 2Asociación Española de Raza Avileña-Negra Ibérica, Padre Tenaguillo 8, 05004 Ávila, Spain; consejoregulador@carnedeavila.org

**Keywords:** maize silage, lipid oxidation, color acceptability, purchase choice

## Abstract

**Simple Summary:**

In recent years, interest has grown in improving the productivity of pure local breeds in Spain using corn silage as an alternative to concentrates. The aim of this investigation was to evaluate the lipid oxidation, color and visual color acceptability of meat from the Avileña-Negra Ibérica breed fed with corn silage in total mixed ration. The inclusion of corn silage led to lower lipid oxidation and higher visual color acceptability when the samples were packaged in a modified atmosphere. The use of corn silage could serve as a feeding option for this type of animal without affecting the quality of the derived meat.

**Abstract:**

In order to increase the economic profitability of Avileña-Negra Ibérica beef production, the inclusion of corn silage in total mixed rations was proposed during the fattening period. Twenty-four Avileña-Negra Ibérica breed entire young bulls were used to evaluate the effects of two feeding systems—concentrate (CON) and corn silage (SIL)—and two packaging systems—vacuum (VAC) and modified atmosphere (MAP)—on the lipid oxidation, color evolution and visual color acceptability of meat throughout display with either oxygen-permeable film (FILM) or modified atmosphere (MAP). Two muscles were assessed: *Longissimus thoracis* (LT) and *Semitendinosus* (ST). Meat from SIL-fed animals had higher TBA values than CON-fed animals after 7 days in MAP packaging. Lipid oxidation was prevented more effectively by VAC packaging than MAP. Display time highly influenced lipid oxidation, since oxidation rates were lower with VAC than MAP packaging. After 14 days of display in MAP, the LT from CON was lighter than the ST. Meat discoloration after 7 days of display was significantly different between packaging systems. Vacuum-packaged meat kept the same color throughout the storage time. Visual color acceptability for the LT was higher throughout display than for the ST. Samples from the MAP were well accepted by consumers, especially the meat from the SIL group. Alternatively, feeding with corn silage could be used for this type of animals without affecting important aspects of meat quality.

## 1. Introduction

Beef producers are interested in offering a variety of meat products. There are factors, such as the animal and its environment, that are linked with the carcass and meat quality [1]. Among these factors, breed and diet are probably the most important [2]. 

The breed of Avileña-Negra Ibérica is found in Central and Southwest Spain. This breed is used exclusively for beef production [3]. Local breed production in Spain is based mainly on semi-extensive or intensive systems. However, increasing the economic returns of beef operations and obtaining differentiated products have become a priority. Because of that, some authors have proposed fattening on pasture supplemented with hay and small amounts of cereals [4], or in fattening units using local by-products and hay [5,6]. However, feeding diets based on silage are also a feasible option, because they can be easily produced in irrigated areas located near fattening units.

In recent years, including maize silage in total mixed rations has roused much interest [5,7,8,9]. Animals finished with corn silage compared with other feeding systems did not show changes or improvements in growth performance or meat quality [5,8,9,10]. Using corn silage reduces the cost of rations by increasing forage consumption without decreasing energy concentration; thus, it could be an alternative for local breeds’ systems. 

The type of packaging used will influence the decision of the consumers when they are able to choose meat from a refrigerated cabinet [11]. The visual appearance, shelf life and eating quality could vary depending on the packaging. Modified atmospheres are commonly used for preserving fresh meat and extending shelf life [12]. A bright red color suggests freshness and superior quality, but in unopened packs, other characteristics such as odor, rancidity or degree of tenderness cannot be assessed. It has been reported that using a gas mixture containing high concentrations of oxygen and low proportions of carbon dioxide provide optimum color stability in red meats [13,14]. On the other hand, refrigeration temperatures have been widely shown to considerably prolong the shelf-life of red meats and delay the growth of spoilage aerobic bacteria [13] as a result of the bacteriostatic effects of carbon dioxide. Several studies have also shown that vacuum conditions are a good option and are a commonly used method for meat preservation [13,15,16].

The color of meat depends on many factors, such as the pH, the characteristics of the muscle, and the concentration and chemical state of the heme pigments, particularly myoglobin. The red color in beef is one of the most important characteristics influencing the consumer’s decision to purchase [17]. The conversion of oxymyoglobin to metmyoglobin produces an unattractive brown color, which equates to the discoloration of the meat [18]. All muscles have a specific physiological purpose resulting in differing fiber types and metabolic function [19]. In consequence, the exposure to atmospheric oxygen provokes unique post-mortem chemistry and color stability in each muscle [20,21,22,23]. Previous reports have shown that meat color stability is muscle-dependent [24], and it is well known that lipid oxidation correlates with heme pigment oxidation in beef [25]. Both types of oxidation are intimately related and are responsible for the smells and flavors of the fat [26]; lipid oxidation is one of the main factors affecting the acceptability of beef.

The aim of the current study was to evaluate the effects of feeding with corn silage vs. those of a concentrate on the color evolution, lipid oxidation and visual acceptability of meat from Avileña-Negra Ibérica young bulls in different packagings throughout display.

## 2. Materials and Methods

### 2.1. Animals and Sampling

In total, 24 Avileña-Negra Ibérica young bulls (250.7 ± 64.7 kg initial live weight and 200.5 ± 44.9 days old), selected randomly from commercial batches of 200 animals, were used to carry out the experiment. One group (*n* = 12) was fed with concentrates and cereal straw ad libitum (CON) and another group (*n* = 12) with a mixture of 70% of corn silage and 30% of concentrate (SIL). For more information about the diets see Barahona et al. [27].

After a period of 250 days, the young bulls (578.7 ± 36.4 kg live weight and 455.2 ± 41.6 days old) were slaughtered using standard procedures and stunning with a captive bolt in an EU-licensed abattoir. Carcasses were chilled at 4 °C for 72 h under commercial conditions. At 72 h post-mortem, the *Longissimus thoracis* (LT) and *Semitendinosus* (ST) muscles were removed from the left side of the carcass, the pH was measured with a penetration electrode Crison, and the meat was vacuum-packaged and sent refrigerated to the University of Zaragoza. At 7 days post-mortem, steaks were sampled. 

### 2.2. Lipid Oxidation

Two 1 cm thick steaks from the LT (T4) and ST (anterior portion) muscles were sliced. One steak from each muscle was divided in two halves that were vacuum-packaged. One of them was kept at 4 °C for 7 days and the other half for 14 days, both in darkness. Another steak was also divided in two halves and packaged with a modified atmosphere (MAP) (70:30 O_2_:CO_2_) with an Ulma Smart 500 machine. All the trays were placed in a Zafrio expositor (Zafrio S.L., Zaragoza, Spain) and kept at 4 °C for 7 or 14 days in simulated retail conditions under light (cool white fluorescent illumination, 1200 lux, 16 h on, 8 off). Lipid oxidation was measured in 10 g meat samples by the 2-thiobarbituric acid (TBA) method [28] with two replicates per animal. The TBA value is an index used for measuring the malondialdehyde (MDA) concentration by the intensity of color developed using a spectrophotometer at 532 nm. The extraction method was based on an extraction with trichloroacetic acid prior to the reaction with TBA. This method compares the absorbance of the MDA-TBA complex with a standard (1,1,3,3 tetramethoxypropano TMP) because the MDA can be obtained by acid hydrolysis from TMP in an equimolecular reaction, so the lipid oxidation was expressed as mg of MDA by kg of muscle.

### 2.3. Color Measurement

Four 3 cm-thick steaks from the LT (T4-T9) and ST (central portion) muscles were obtained. Two of them were vacuum packaged and kept at 4 °C for 7 and 14 days in darkness. The other two steaks were kept under MAP (70:30 O_2_:CO_2_) at 4 ^o^C for 7 and 14 days of blooming under simulated display, as has been indicated previously. Color measurements were taken immediately when the steaks were removed from the muscles (point 0), and after 7 and 14 days of packaging either in the dark or under MAP, using a Minolta CM2002 Spectrophotometer in the CIEL*a*b* space [29]. Hue (H°), chroma (C*) and discoloration (Dec, R630-R580) were calculated to estimate the degree of meat discoloration [30]. All the measurements were performed immediately after opening the packaging as the average of three measures in the surface of the steak.

### 2.4. Visual Color Acceptability

Two 2 cm thick steaks per animal from the LT (T9-T11) and ST (posterior portion) were cut, vacuum packaged, aged for 7 days, frozen and kept at −18 °C. The steaks were thawed at 4 °C for 24 h before the analysis. One of the steaks from each muscle was placed in a polystyrene tray overwrapped with an oxygen-permeable film (O_2_ transmission rate of 650–750 cm^3^/m^2^ day at 25 °C and 0% relative humidity; Irma S.A., Zaragoza, Spain), and the other was stored in MAP (70:30 O_2_:CO_2_). A set with 4 fluorescent tubes and a fixed camera (Olympus P9000) was set up for taking photos under similar constant conditions. A picture was taken of each tray daily. Each tray was identified with a 2-digit code that was randomly changed every day to avoid sample recognition. Fifteen consumers (from staff at the university) assessed the visual color acceptability (from 1, extreme dislike, to 9, extreme like) and the purchase choice (yes or no) of all samples from the pictures, across 24 sessions (12 sessions per muscle) on the same computer with the same screen and light, at the subject’s convenience. Each consumer completed the evaluation on their own. Consumers evaluated 48 photos in each session in no more than 2 sessions per day. All the consumers started with the samples from LT, and moved from the first day of display to the last one. After that, they did the same with samples from ST.

### 2.5. Statistical Analysis

All statistics were calculated using the SPSS 22.0 statistical package. TBA and color measurements were analyzed by the GLM procedure considering feeding, muscle, packaging and display or ageing, and their interactions were used as fixed effects. Duncan’s multiple range test with a significance of *p* < 0.05 was used to assess differences between means. 

Data on visual color acceptability were analyzed by analysis of variance with repeated-measures ANOVA and Kolmogorov–Smirnov tests for the normality of data distribution. Feeding and packaging were used as the fixed effects. 

Data from purchase were analyzed by frequencies and the chi-square test was used for significant differences between frequencies from different days of analysis.

## 3. Results and Discussion

Results were analyzed first as a pull, studying the effects of feeding, muscle, packaging and display on all the variables and their respective interactions. However, because of the number of significant interactions between packaging and the rest of the effects, and in order to understand better the behavior of the two muscles and the effect of diet, the results have been separated by packaging. 

### 3.1. Lipid Oxidation

The effects of feeding, muscle and display, and their interactions, on lipid oxidation are shown in Table 1. Feeding had a significant influence on lipid oxidation when samples were stored under vacuum conditions (*p* = 0.045), but did not significantly affect it when the samples were stored under MAP (*p* = 0.057), probably because the high presence of oxygen in MAP had the largest influence on lipid oxidation throughout display (*p* < 0.001), masking other effects. However, a significant interaction was observed between feeding and display. When samples were stored under vacuum conditions, meat from SIL-fed groups showed higher lipid oxidation (*p* = 0.036) at 14 days of storage (Table 2). Although feeding did not influence oxidation in MAP (*p* = 0.057), meat at 7 days of display from CON showed higher lipid oxidation than meat from the SIL group (0.970 vs. 0.672 mg of MDA/kg of meat, respectively). Other authors have reported lower oxidation in meat from grass silage-fed animals with higher vitamin E concentrations when compared to meat from maize-fed animals. [31,32]. The use of maize silage could preserve the meat from lipid oxidation more effectively than meat from concentrate-fed animals. However, it is probable that a higher proportion of PUFA could increase the lipid oxidation, especially during storage [33]. In both types of packaging, lipid oxidation did not differ between muscles (Table 2).

### 3.2. Color

The lipid oxidation values of meat stored in MAP packaging were higher than those for meat samples stored in vacuum, and reached 2 mg of MDA/kg of meat (Table 2), which is considered the limiting threshold for acceptability [34]. This threshold was reached at 7 days in MAP, but was not reached at all in vacuum-packaged meat. Other researchers have reported an increase in oxidation throughout the display of fresh beef [35]. Modified-atmosphere packaging resulted in more oxidation than vacuum packaging, limiting the shelf life of MAP packaging [11], as expected considering that the higher level of oxygen and the prooxidation effect of the light during display boosted lipid oxidation [36].

Feed significantly affected lightness (L*), redness (a*), chroma (C*) and hue (H°) when the meat was stored in vacuum packaging, but did not affect yellowness (b*) or discoloration (DEC). Furthermore, muscle type affected all color parameters except redness and chroma. Packaging meat in a modified atmosphere had a greater effect on color, and masked any effect that might be derived from the feed. Meanwhile, muscle type affected the rate of discoloration, but none of the other color variables (Table 1). 

Figure 1 shows the significant interactions between muscle and packaging conditions. Meat from animals fed SIL was lighter (L*) than meat from CON-fed animals when stored in vacuum packaging. These results contradict those obtained by Avilés et al. [2], where the meat was darker from animals fed on total mixed ration than from animals finished on concentrate. The ST muscle presented higher luminosity than samples from LT, independently of the packaging, probably because of the oxidative activity of the semitendinosus, having less color stability than the LT [37,38]. Lightness increased as the time of display increased, especially in MAP, due to the higher availability of oxygen (Figure 1).

Feeding (*p* < 0.001) and display (*p* < 0.05) had a significant effect on redness (a*) when samples were stored in VAC (Table 1). Samples from animals fed CON were redder than samples from animals fed SIL, especially after 7 days of storage (Figure 1). When samples were stored in MAP packaging, both muscle and display time affected (*p* < 0.001) redness (Figure 1). The ST muscle had a similar red color from 0 to 7 days, while that of the LT increased up to 7 days of storage, and then both muscles showed a decline in redness up to 14 days of display. At 14 days of display, MAP packaging resulted in meat with lower red values than VAC packaging, independently of the type of feeding. It is known that oxymyoglobin increases during the first days of storage in MAP because of the presence of oxygen; however, longer display times induce the formation of MetMb, and thus lower color stability, during storage in MAP [39]. Meat color stability is the net result of autoxidation and the reduction of myoglobin. It is known that in the post-mortem period, the MetMb-reducing activity decreases [40,41,42], which could explain the lowest redness of meat after 14 days of storage in MAP. 

Yellowness values (b*) were affected by muscle and display in both packagings, but not by feeding (Table 1), even though silage might include a higher content of carotenoids that could have been incorporated into the intramuscular fat, changing the color towards more yellow notes [43]. In older animals, this effect would probably have been more evident. The ST muscle was more yellow than the LT muscle in both packaging systems. However, the behaviors throughout display were different, since MAP meat had higher values of yellowness than VAC meat, especially at 7 days of display (Figure 1). 

Chroma was affected by feeding and muscle in VAC-packaged meat (Figure 2). Meat from animals finished on concentrates had higher chroma values than meat from animals fed silage. Furthermore, chroma values were higher for ST than LT. The chroma values were influenced by muscle and display time, but not feeding, in meat stored in MAP packaging. At 0 and 14 days of display, samples from the ST muscle presented higher chroma values than LT; however, at 7 days of display, these differences were not shown. Samples in MAP had chroma values higher than samples in vacuum packaging, but after 14 days of display these differences disappeared. This is the consequence of the faster oxygenation rate in samples surrounded by a high oxygen atmosphere, as happens in MAP, and the formation of metmyoglobin (MMb) afterwards. This MMb formation also happens under vacuum conditions, but the residual oxygen in the packaging cannot stimulate the formation of oxymioglobin at the same rate as happens in MAP [44]. At 7 days of display, samples from both muscles in MAP had chroma values over 18. Chroma values above 18 have been reported as acceptable to consumers [45]. This suggested threshold was based on older animals than the yearlings used in the present study and meat from grass-fed animals. Nevertheless, our results are in agreement with those of Casasús et al. [5], who also found that corn silage-fed yearlings had chroma values between 20 and 18 during the 13 days of display. Some authors have reported acceptable chroma values lower than 18 [46] in young animals, which do not show the degree of color saturation that old or grass-fed animals can reach. Therefore, VAC samples could have chroma values that are also acceptable for consumers.

Hue values were affected by muscle and display time in both VAC and MAP packaging, while feed effect was only significant in samples that were vacuum-packaged (Table 1). Samples from corn silage-fed animals showed higher hue values, especially in the LT muscle (Figure 2). Under VAC, hue increased from day 0 to day 7, and after that stayed stable up to 14 days of display. Meat from the ST muscle had higher hue values than meat from the LT muscle, no matter what packaging system was used. A partial explanation for this could be related to the different fiber types in each muscle, with the ST having more fast-switch fibers that would increase the hue [23]. Besides, some authors have suggested that the ST muscle has greater lightness and hue, but less redness [47,48]. Under MAP, the hue angle increased from day 0 to day 14, although that increment was more pronounced in ST than LT.

In samples that had been vacuum-packed, discoloration was significantly affected by muscle (Table 1). Discoloration occurred faster in the ST muscle than the LT. The VAC system resulted in little discoloration from 0 to 14 days of storage. Feeding had no effect (Table 1). Without subjecting samples to display, Wales et al. [49] also reported that increasing the maize silage proportion of a concentrated ration of dry matter from 46% to 96% did not result in a difference either in muscle color or in fat. Similarly, Walsh et al. [50] observed no differences in fat or muscle color when comparing two rations containing 74% of maize silage and 90% concentrate. On average, maize silage appears to have a similar effect to concentrate on the nutritional quality of beef [43]. 

As shown in Table 1, when MAP packaging was used, significant differences related to the muscle disappeared, and the DEC decreased as time of display increased (Figure 3). Although modified-atmosphere packaging (MAP) is used to present meat in a more attractive way for the consumer, the shelf life of beef in MAP is limited when it has been previously stored under vacuum conditions, due to the subsequent decrease in color stability during display [20,51,52]. The decrease was more pronounced in ST than LT muscle, because in LT, between 0 and 7 days, there were no differences and no degradations in the color. This might be the effect of the different fiber compositions of these two muscles, with a higher number of white fibers that are more glycolytic and hence more unstable in ST than in LD. Some authors have also reported that muscles such as LT have a more stable color than other muscles such as ST [37,38].

The consumer often tends to associate color with several attributes, such as flavor, tenderness, safety, storage time, nutritional value and satisfaction level [53]. Color allows the detection of certain anomalies or defects that food items may present [54,55,56], and redness in particular is used as an indicator of freshness [57]. 

### 3.3. Visual Color Acceptability and Purchase Choice

Visual color acceptability was influenced (*p* < 0.001) by feeding, muscle and display, independently of the packaging used in the display—MAP or FILM. As is shown in Table 3, meat from the ST of CON-fed animals and kept in FILM packaging presented better color acceptability than meat from the LT muscle during the first 3 days. After that, the LT muscle showed higher scores than ST. In the case of meat from animals fed with corn silage, the better color acceptability of the LT muscle occurred after 2 days of display. In both types of feeding, the LT muscle presented better visual color acceptability throughout display, the scores being above 5 until day 7 of display. In the case of ST, after 4 days of display the scores were below 5. Thus, the LT muscle in FILM packaging had better acceptability than the ST muscle.

The samples in MAP kept their visual color acceptability longer than samples in FILM (Table 4). Although in the first 3 days, the ST muscle from the SIL group showed better acceptability, at 4 days of display, the LT presented higher scores. The same happened in samples from the CON group after 4 days of display. Besides this, the visual color acceptability of the LT muscle lasted longer than that of the ST muscle, especially in samples from the SIL group. Meat from the LT muscle from SIL animals had scores over 5 until day 7 of display (5.48). In general, the use of modified-atmosphere packaging could preserve the color acceptability of beef more effectively than FILM. Thus, the SIL treatment was the most accepted in the LT muscle. 

The percentages of positive assessments of buying meat from the FILM and MAP groups are presented in Table 5 and Table 6, respectively. As occurred with the visual color acceptability, in the case of the SIL group, in the first 2 days of display, the percentage of positive assessments of buying the meat was slightly higher in the ST muscle group than in the LT muscle group. However, at 3 days of display, meat from LT had a higher percentage of purchase intention than ST, and this behavior was maintained during the display. In FILM, the LT and ST samples from SIL-fed animals received 77.2% and 6.1% positive assessments of intention to buy at 6 days of display, respectively. LT and ST samples from CON-fed animals after 6 days of display had positive assessments of intention to buy of 50% and 2.3%, respectively. In the case of samples kept in FILM, the decrease was pronounced, especially in ST. At 5 days of display, the percentages of intention to purchase were 32.8% and 35.0% for CON and SIL, respectively, and at 6 days the percentages went down to 2.3% (CON) and 6.1% (SIL).

Meat in kept MAP packaging for 8 days of display had almost 50% positive assessments in both the CON and SIL groups of LT muscle (45.6% and 55.0%, respectively). However, MAP-packaged ST had less than 30% purchase choice after 7 days of display (CON: 28.3% and SIL: 24.4%). One of the most important characteristics of meat is the color. This attribute influences the acceptability of the product and plays a major role in the purchase decision [44,58,59]. The greater deterioration of ST vs. LT implies the lower purchase acceptability of ST. Carpenter et al. [60] found a strong relationship between color preferences and consumer purchasing decisions, as consumers discriminated against beef that was not red (e.g., purple or brown). Therefore, visual assessments are a gold standard for estimating consumer perception [44].

## 4. Conclusions

We can recommend the use of corn silage for commercial meat from Avileña-Negra Ibérica breed animals, especially for muscles of a higher category, such as longissimus thoracis, without decreasing quality parameters such as color. However, special attention has to be paid after 7 days of display if this commercialization is performed in modified-atmosphere packaging instead of vacuum packaging, due to the increased lipid oxidation and color deterioration. 

The type of muscle can affect the acceptability of the color of the meat. Nevertheless, the type of packaging is an important factor to be taken into account, because modified-atmosphere packaging preserves meat color better than oxygen-permeable film.

## Figures and Tables

**Figure 1 animals-11-02863-f001:**
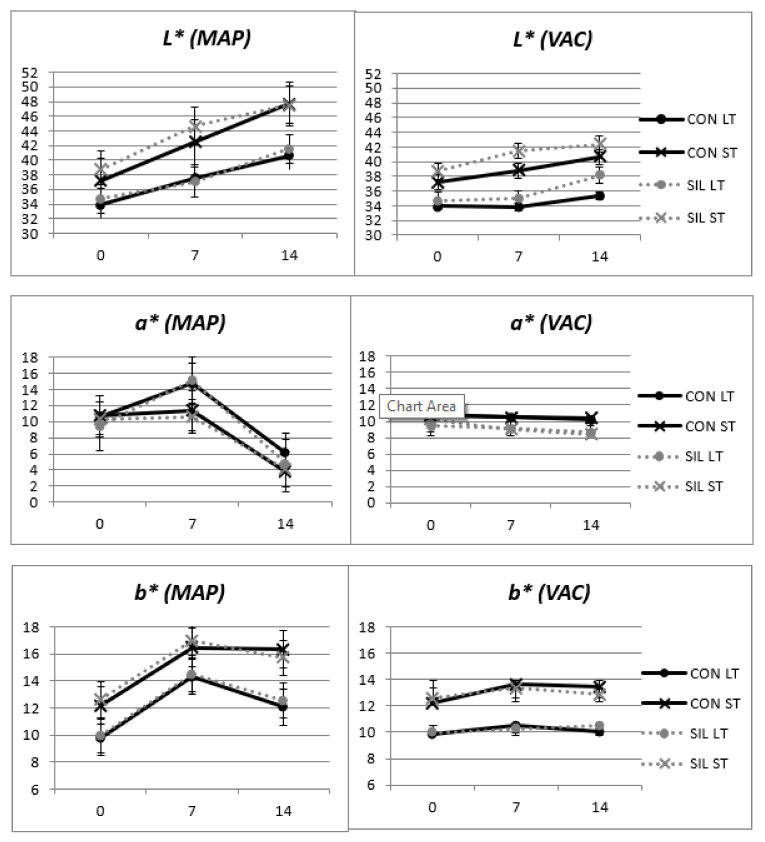
Effect of feeding and type of muscle on the luminosity (*L**), redness (*a**) and yellowness (*b**) of LT and ST samples from the Avileña-Negra Ibérica breed kept in modified-atmosphere packaging (MAP) and a vacuum (VAC). CON: concentrate; SIL: maize silage; LT: *longissimus thoracis*; ST: *semitendinosus.*

**Figure 2 animals-11-02863-f002:**
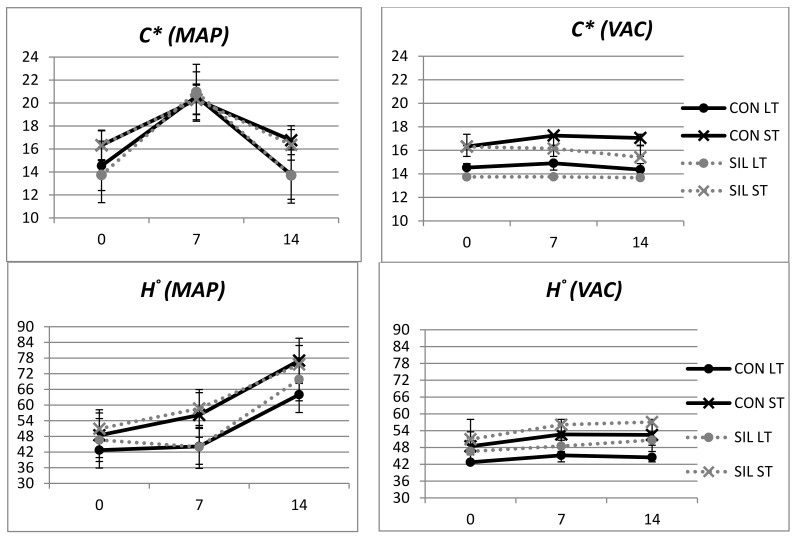
Effect of feeding and type of muscle on luminosity (*L**), redness (*a**) and yellowness (*b**) of LT and ST of modified-atmosphere packaging (MAP) and vacuum (VAC) samples of meat from Avileña-Negra Ibérica animals. CON: concentrate; SIL: maize silage; VAC: vacuum packaging; MAP: modified-atmosphere packaging. LT: *longissimus thoracis*; ST: *semitendinosus.*

**Figure 3 animals-11-02863-f003:**
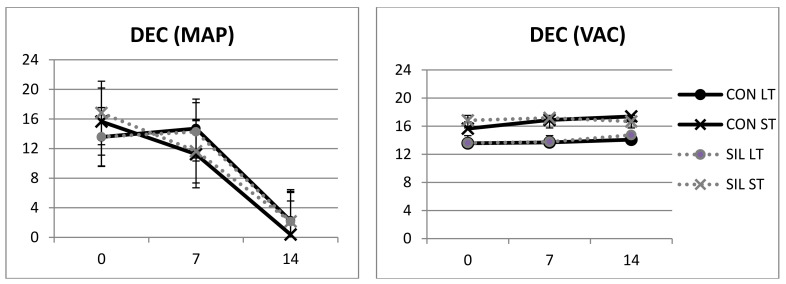
Effect of feeding and type of packaging on R630-580 of modified-atmosphere packaging (MAP) and vacuum (VAC) samples of meat from the Avileña-Negra Ibérica breed. CON: concentrate; SIL: maize silage; VAC: vacuum packaging; MAP: modified-atmosphere packaging. LT: longissimus thoracis; ST: semitendinosus.

**Table 1 animals-11-02863-t001:** *p*-value of feeding (FEED), muscle (MUS), display (DIS) and their interactions on lipid oxidation (TBA), luminosity (L*), redness (a*) yellowness (b*), chroma (C*), hue (H°), discoloration (R630-580) (Dec) and visual color acceptability (VCA) on meat from Avileña-Negra Ibérica breed, in two different packaging systems: vacuum (VAC) and modified atmosphere (MAP).

	VAC						MAP					
	FEED	MUS	DIS	FxM	FxD	MxD	FEED	MUS	DIS	FxM	FxD	MxD
**TBA**	0.045	0.649	0.111	0.053	0.020	0.628	0.057	0.184	<0.001	0.408	0.311	0.666
**L***	<0.001	<0.001	<0.001	0.658	0.527	0.084	0.058	<0.001	<0.001	0.349	0.740	0.007
**a***	<0.001	0.585	0.017	0.895	0.312	0.583	0.192	<0.001	<0.001	0.581	0.787	<0.001
**b***	0.983	<0.001	0.014	0.516	0.630	0.526	0.398	<0.001	<0.001	0.723	0.757	0.039
**C***	0.002	<0.001	0.525	0.941	0.489	0.957	0.628	<0.001	<0.001	0.967	0.689	<0.001
**H**°	<0.001	<0.001	<0.001	0.478	0.312	0.218	0.068	<0.001	<0.001	0.395	0.777	0.017
**Dec**	0.331	<0.001	0.038	0.987	0.599	0.480	0.464	0.499	<0.001	0.313	0.856	0.002
**VCA**	<0.001	<0.001	<0.001	<0.001	<0.001	<0.001	<0.001	<0.001	<0.001	0.007	<0.001	<0.001

FxM: interaction FEEDxMUSCLE; FxD: interaction FEEDxDISPLAY; MxD: interaction MUSCLExDISPLAY. Interaction FEEDxMUSxDIS not significant.

**Table 2 animals-11-02863-t002:** Effect of the feeding (concentrate vs. maize silage) and used muscle (*Longissimus thoracis* vs. *Semitendinosus*) of Avileña-Negra Ibérica breed on lipid oxidation (TBA) (mg of malonaldehyde/kg of meat) during display (0, 7 and 14 days) in two different systems of packaging: vacuum (VAC) and modified atmosphere (MAP).

VAC									MAP							
	FEED		MUSCLE			*p*-Value			FEED		MUSCLE			*p*-Value		
Display	CON	SIL	LT	ST	SEM	FEED	MUS	FxM	CON	SIL	LT	ST	SEM	FEED	MUS	FxM
0	0.081	0.089 B	0.082	0.088	0.005	0.451	0.629	0.017	0.081 C	0.089 C	0.082 C	0.088 C	0.005	0.451	0.629	0.017
7	0.089	0.082 B	0.089	0.082	0.003	0.197	0.145	0.073	0.970 aB	0.672 bB	0.733 B	0.896 B	0.065	0.018	0.203	0.243
14	0.078 b	0.138 aA	0.101	0.118	0.014	0.037	0.563	0.409	2.244 A	2.059 A	2.062 A	2.223 A	0.106	0.394	0.438	0.871
SEM	0.002	0.010	0.006	0.009					0.119	0.117	0.119	0.118				
*p*-Value	0.182	0.036	0.458	0.194					<0.001	<0.001	<0.001	<0.001				

VAC: vacuum packaging; MAP: modified-atmosphere packaging; CON: concentrate; SIL: maize silage; LT: longissimus thoracis; ST: semitendinosus. MUS: muscle FxM: interaction FEEDxMUSCLE. Means with different letters are significantly different within row and effect (*p* < 0.05). Means with different capital letters are significantly different within column.

**Table 3 animals-11-02863-t003:** Effect of feeding (concentrate vs. maize silage), muscle (longissimus thoracis vs. semitendinosus) and display on visual color acceptability of samples kept in oxygen-permeable film packaging (FILM) of the Avileña-Negra Ibérica breed.

	CON		SIL			*p*-Value		
Days	LT	ST	LT	ST	SEM	FEED	MUS	FEEDxMUS
1	6.98 c	7.57 ab	7.25 bc	7.65 a	0.060	0.129	<0.001	0.428
2	7.05 b	7.28 ab	7.39 a	7.43 a	0.055	0.024	0.229	0.426
3	6.36 b	6.46 b	6.97 a	6.59 b	0.056	0.001	0.219	0.032
4	5.99 b	5.31 c	6.55 a	5.60 c	0.058	<0.001	<0.001	0.234
5	5.39 b	4.46 c	5.93 a	4.57 c	0.058	0.002	<0.001	0.046
6	5.19 b	3.41 d	6.07 a	3.71 c	0.066	<0.001	<0.001	0.004
7	3.64 b	2.55 c	4.32 a	2.69 c	0.061	<0.001	<0.001	0.014
8	2.99 b	1.71 c	3.51 a	1.93 c	0.048	<0.001	<0.001	0.062
9	2.19 b	1.26 c	2.67 a	1.32 c	0.041	<0.001	<0.001	0.003
10	1.59 b	1.00 c	1.96 a	1.00 c	0.029	<0.001	<0.001	<0.001
11	1.24 a	1.00 b	1.27 a	1.00 b	0.013	0.519	<0.001	0.519
12	1.00	1.00	1.00	1.00	0.000	ns	ns	ns

CON: concentrate; SIL: maize silage; FEED: feeding; LT: longissimus thoracis; ST: semitendinosus. Means with different letters within a row are significantly different (*p* < 0.05).

**Table 4 animals-11-02863-t004:** Effect of feeding (concentrate vs. maize silage), muscle (longissimus thoracis vs. semitendinosus) and display on visual color acceptability of samples from modified-atmosphere packaging (MAP) of Avileña-Negra Ibérica breed meat.

	CON		SIL			*p*-Value		
Days	LT	ST	LT	ST	SEM	FEED	MUS	FEEDxMUS
1	7.35 b	7.78 a	7.58 ab	7.67 ab	0.059	0.600	0.024	0.144
2	6.88 b	7.76 a	7.19 b	7.70 a	0.062	0.285	<0.001	0.122
3	6.57 b	7.07 a	6.74 b	7.29 a	0.059	0.091	<0.001	0.818
4	6.72 b	6.47 b	7.02 a	6.72 b	0.052	0.007	0.008	0.823
5	5.85 b	5.49 c	6.32 a	5.88 b	0.052	<0.001	<0.001	0.693
6	6.12 a	4.67 c	5.82 b	4.90 c	0.057	0.775	<0.001	0.012
7	4.58 b	3.76 c	5.48 a	3.94 c	0.065	<0.001	<0.001	0.003
8	3.81 b	3.08 c	4.49 a	3.37 c	0.062	<0.001	<0.001	0.092
9	3.15 b	2.61 c	3.78 a	2.89 b	0.049	<0.001	<0.001	0.087
10	3.02 a	2.38 b	3.27 a	2.51 b	0.047	0.056	<0.001	0.521
11	2.64 b	1.91 d	2.88 a	2.17 c	0.041	0.002	<0.001	0.896
12	2.47 a	1.00 b	2.57 a	1.00 b	0.037	0.403	<0.001	0.403

CON: concentrate; SIL: maize silage; FEED: feeding; LT: longissimus thoracis; ST: semitendinosus. Means with different letters within a row are significantly different (*p* < 0.05).

**Table 5 animals-11-02863-t005:** Percentage (%) of assessments for purchasing samples from permeable oxygen film (FILM) throughout display.

	CON		SIL	
Days	LT	ST	LT	ST
1	91.7 a	93.9 a	91.1 ab	96.7 a
2	90.6 a	93.9 a	93.3 b	94.4 ab
3	75.6 b	83.3 a	87.2 abc	85.6 b
4	68.3 bc	49.4 b	81.7 abc	68.3 c
5	60.6 bc	32.8 b	79.4 ac	35.0 d
6	50.0 c	2.3 c	77.2 c	6.1 e
7	21.1 d	0.6 c	30.0 d	1.1 ef
8	7.8 d	0 c	11.7 e	0 f
9	7.8 d	0 c	7.8 e	0 f
10	0 e	0 c	0 f	0 f
11	0 e	0 c	0 f	0 f
12	0 e	0 c	0 f	0 f

CON: concentrate; SIL: silage. Percentages with different letters within a column are significantly different (*p* < 0.05).

**Table 6 animals-11-02863-t006:** Percentage (%) of assessments for purchasing samples from modified-atmosphere packaging (MAP) throughout display.

	CON		SIL	
Days	LT	ST	LT	ST
1	88.9 a	95.6 a	92.8 a	97.2 a
2	88.3 a	97.2 a	93.9 a	96.1 a
3	88.9 a	96.1 a	86.1 ab	95.6 ab
4	90.6 a	92.8 a	93.9 a	90.0 ab
5	86.7 a	67.2 b	91.1 bc	85.6 c
6	84.4 a	45.6 c	76.7 cd	58.3 c
7	54.4 b	28.3 cd	70.0 d	24.4 d
8	45.6 b	13.9 d	55.0 e	13.9 d
9	18.9 c	11.7 d	29.4 e	16.1 d
10	19.4 cd	18.3 d	23.9 e	20.0 d
11	15.0 cd	18.9 d	16.1 ef	18.3 d
12	0 d	0 e	0 f	0 e

CON: concentrate; SIL: silage. Percentages with different letters within a column are significantly different (*p* < 0.05).

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
