# Peer review of "Feeding, Muscle and Packaging Effects on Lipid Oxidation and Color of Avileña Negra-Ibérica Beef"

_animals, 2021, doi:10.3390/ani11102863_

Round 1
Reviewer 1 Report
Please see attached file.

Author Response
REVIEWER 1
Line 25-26 Meat from SIL fed animals had higher TBA values than CON fed animals…..
Thanks for your comment, we have changed it
Line 27 Display time highly influenced lipid oxidation
Done
Line 28 – 29 Problem with this sentence suggest - Meat discoloration after 7 days of display
was significantly different between packaging systems. Vacuum packaged meat had the same
colour throughout the storage time.
We have changed it
Line 29-30 Visual colour acceptability for the LT was higher throughout display than for ST.
Ok, it´s done
Line 64-65 Sentence awkward. Consider rewording.
We reworded that sentence for better understanding.
Line 69 switch meat with beef Cherry red colour is beef not the more generic term meat.
Ok, we changed it.
Line 100 – 102 Was the gas measured from the package to verify the composition?
The composition is measured as a routine practice from an empty package, the same as the calibration of the spectrophotometer.
Line 118-122 How many measurements were made on each steak? If working to identify discoloration it would seem that you need more than one measurement as all areas of the steak may or may not discolor at the same rate. Also, if measuring directly from the package, the vacuum packaged product would be in the myoglobin state not in the oxymyoglobin state. This seems to go against the objective of a bright cherry red cut.
We made 3 different measures on the surface of the steak for each single observation. This has been added in the text (L124-125).
Line 123 Why use a 2 cm thick steak for visual evaluation and 3 cm for the objective color measurements? Also, seems odd to freeze the color sample for visual evaluation. There is an effect of freezing on color. How were the steaks packaged for frozen storage?
To measure color with a spectrophotometer, a 3-cm thickness is required in order not to lose light through the sample if too thin. However, there is no need for so much thickness for visual assessments since the human eye can only see the surface. The steaks were vaccum packaged, and then aged for 7 days in a fridge (±4°C) and finally kept at -18°C for technical reasons to do photographs at the same time. Color can be altered, as the reviewer indicates, but all treatments were affected at the same rate. Visual assessments with photographs has previously been reported with success (Cortez Pasetti et al., 2017. DOI: 10.1016/j.meatsci.2016.09.009; Cortez Pasetti et al., 2019. DOI: 10.1016/j.foodres.2019.03.036)
133 – 136 Sentence is awkward. Please revise. I am also a little confused on how the visual evaluation was conducted. Was there some confirmation that the color for the pictures were true?
All the pictures were made in the same conditions. Previous studies performed in our group (see previous answer)indicates the possibility of performing acceptability test though pictures, so that the same image is assessed by many people, without modifying its characteristics. All consumers performed the analysis in the same computer, with the same screen and light. This has been added to the text (L138-139).
Line 137 – 138 Consumers evaluated 48 photos in each session with no more than 2 sessions per day.
OK
Line 138 – 139 Why not a random order for the photos. If you did first day to last there will be some bias associated with the measurements.
We have previously demonstrated that random presentation results in lower scores and slower acceptability decrease than sequential presentation, but only between the 2nd and 5th days of display (Cortez Pasetti et al., 2017). With the large amount of photographs that consumers evaluated at the end of the trial, in a large period of time and unable to compare from one day to the following one, we believe that the bias was minimum. All treatments were evaluated each day, so that the comparison between treatments was not affected by the date of evaluation.
Line 164 – 165 When samples were stored under vacuum conditions, meat from SIL fed groups
showed higher lipid oxidation at 14 days of storage.
Ok, we have changed it.
Line 168 – 171 Other authors have reported lower oxidation in meat from grass silage fed animals with higher vitamin E concentrations when compared to meat from maize fed animals.
Ok, it is done.
Line 176 – 179 Lipid oxidation values of meat stored in MAP packaging were higher than meat samples stored in vacuum and reached 2 mg of MDA/kg of meat considered the limiting threshold for acceptability. This threshold was reached after 7 days in MAP and was not reached at all in vacuum packaged meat.
Ok, we have changed it.
Line 179-180 Other researchers have reported an increase in oxidation throughout display of
fresh beef.
Ok
Line 181 – 183 Modified atmosphere packaging resulted in more oxidation than vacuum packaging, limiting the shelf life of MAP packaging. The higher level of oxygen and the prooxidation effect of light during display boosted lipid oxidation.
Ok
Table 2 very difficult to read need more space between columns.
We have changed the space between columns.
Line 196-201 Feed significantly affected lightness, redness, chroma and hue when the meat was stored in vacuum packaging but did not affect yellowness or discoloration. Furthermore, muscle type affected all colour parameters except redness and Chroma. Packaging meat in a modified atmosphere had a greater effect on colour and masked any effect that might be from feed. Meanwhile, muscle type affected rate of discoloration but none of the other color variables (Table 1)
Ok, it is done.
Line 202- 204 Figure 1 shows the significant interactions between muscle and packaging conditions. Meat from animals fed SIL was lighter (L*) than meat from CON fed animals when stored in vacuum package.
Ok, we have changed it.
Line 204 – 206 Were the samples packaged the same? Was there information about pH? Both of these could explain the difference reported here. Might want to expand this part.
Yes, samples were subjected to the same packaging conditions. There were no abnormal pHs. This has been added to the text (L95-96)
Line 210 (Fig. 1) You have figure in parenthesis elsewhere.
Ok
Line 211 -212. Feeding (P<0.001) and display (P<0.05) had a significant effect on redness when samples were stored in vacuum package (Table 1).
Ok, it´s done.
Line 212 – 213 Samples from animals fed CON were reader than samples from animals fed SIL, especially after 7 days of storage (Figure 1).
Ok, we have changed it.
Line 213 - 214 When samples were stored in MAP packaging, both muscle and display time effected (P<0.001) redness (Figure 1). The ST muscle had similar red color from 0 to 7 days while the LT increased in redness up to 7 days of storage and then both muscles had a decline in redness to 14 days of display.
Ok, we have changed it.
Line 216 -217 After 14 days of display, MAP packaging was less red than samples VAC packaging, independent of feeding regime. You suggest this, but it depends on the display day. If you have look at the different display times, the sample is the same at 0 days, the VAC samples for CON LT was redder at 7 days and then all MAP samples were less red at 14 days. With all of the interaction, the general statement of MAP packaging was less red is problematic.
In this case, there was a significant interaction between display and muscle, but only in MAP, not in VAC samples, and due to this interaction, we showed the data in the graph by individual treatment. As it can be seen in Fig1, all samples in MAP are below 6 in a* value, whereas in VAC all values are above 8. Therefore, we do not see problematic the sentence, since we believe it is what happens at long display times (L221-222).
Line 219 – 221. “display time” is more appropriate than aging time for colour ….There is a problem with the rest of this sentence. I think you are saying there is an increase in metmyoglobin no matter the packaging. However, if a vacuum package is completely sealed you have no variation in the colour nor development of metmyoglobin, thus the flat line for a* value.
We agree with the reviewer with this comment. Display time has been corrected and so has the sentence, that was wrongly expressed, since MAP shows lower stability at long display time than vacuum packaging. (L224-225).
Line 245 – 247 Chroma was affected by feeding and muscle in VAC meat (Figure 2). Meat from animals finished on concentrate had higher chroma values than meat from animals fed silage. Furthermore Chroma values were higher for ST than LT.
OK
Line 247 - 248 Chroma values were influenced by muscle and display time but not feeding in meat stored in MAP packaging.
OK
Line 254-255 Metmyoglobin forms in the vacuum package early during the packaging. It should have all went to deoxymyoglobin before you got past 2-3 days even if you had the display temperature at 0C. The statement you have doesn’t fit the way you designed the study. Unless there is something missing from your materials and methods.
We have not measured the amount of metmyoglobin. This sentence is part of a discussion based on other authors’ findings (Mancini and Hunt, 2005). We agree with the reviewer since our design is not orientated to this conclusion. (L257-261).
Line 256 -257 AT 7 days of display, samples from both muscles in MAP had Chroma values over 18. Chroma values above 18 have been reported as acceptable to consumers.
Ok, we have changed this sentence.
Line 265-267 Hue values were affected by muscle and display time in both VAC and MAP packaging while feed effect was only significant in samples that were vacuum packaged (Table 1). Samples from corn silage fed animals showed higher hue values especially in LT muscle (Figure 2).
We have changed it
Line 268 – 271 ST muscle had higher Hue values than LT muscle, no matter the packaging system. A partial explanation for this could be related to the different fiber types in each muscle with the ST having more fast twitch fibres that would increase the Hue.
Ok, we have changed this sentence.
Line 280 – 282 In samples that had been VAC, discoloration was significantly affected by muscle and display time (Table 1). Discoloration occurred faster in ST muscle than in LT. VAC system resulted in little discoloration from 0 to 14 days of storage.
Ok, we have changed this sentence.
Line 283 – 287 Did the references you used look at storage time? The way you have this presented it suggests this is about absolute color of fat and lean not necessarily over time. Does this fit in the discoloration section?
No, these authors only measured at the moment of sampling. We used these references as a way to justify our lack of differences due to feeding, since other authors have also found the same. We have added to the text that this lack of differences were found without display (L289-290)
Line 321 – 322 Please reword.
Done (L328-330)
Line 345 – 348 LT and ST samples from SIL fed animals had 77.2% and 6.1% positive assessments to buy the samples at 6 days of display. LT and ST samples from CON fed animals, after 6 days of display, had positive assessments to buy at 50% and 2.3%, respectively.
Ok, we have changed this sentence.
Line 356 – 358 Meat in MAP packaging for 8 days of display had almost 50% of assessments positive for both CON and SIL group on LT muscle (45.6% and 55% respectively). However, MAP packaged ST had less than 30% purchase choice after 7 days of display (CON 28.3% and SIL 24.4%).
Ok, done.
Conclusions are awkward. Please reword.
Conclusions have been reworded.

Reviewer 2 Report
Manuscript animals-1271121, entitled “Feeding, muscle and packaging effects on lipid oxidation and colour of Avileña Negra-Ibérica beef”
Recommendation: The above paper is not suitable for publication in its present form.
General comment
The article provides useful information about the effects of diet, muscle type and packaging on lipid oxidation and colour of Avileña Negra-Ibérica beef. Although, the experiment is in general appropriately designed and implemented, there are some points that should be corrected or clarified.
Major comments
- Please use uniform description for vacuum packaging. VAC or FILM?
- Why visual color acceptability was performed through photographs?
- In Table 2, please let a space among columns or use two instead of three decimals
- Please explain what is “the percentage of positive assessments” in Material and Methods
- Please merge paragraph L356-365 with that in L324-329 (repetition).
Minor points:
L17: “serve as” instead of “be”
L18: “…of the derived meat.”
L19: “profitability” instead of “returns”
L24-25: Please delete “throughout display with either oxygen permeable film (FILM) or modified atmosphere (MAP)”
L26-27: “Display highly influenced lipid oxidation, since oxidation rates were lower by VAC than MAP packaging.”
L28-29: What do you mean? Please rephrase
L37: “a variety of meat” instead of “differentiated”
L40: “The local breed of Avileña-Negra Ibérica is located in…”
L46: “…are also a feasible…”
L182: “considering” instead of “given”
L199: Please rephrase
L214: “…was also affected by…”
L217: “resulted in meat with lower red value” instead of “showed meat less red”
L245: “…in VAC packaged meat. As it…”
L247: “…the inclusion of silage, with these values being higher in ST than LT muscle. Chroma value of MAP packaged meat was…”
L249: “…presented higher Chroma values than LT…”
L250: “were not shown” instead of “disappeared”
L265-266: “…at any packaging, with feeding being also significant under VAC…”
L276: “Figure 2.”
L280: “affected” instead of “different”
L283: “Table 1” instead of “Figure 3”
L332: VAC?
Author Response
REVIEWER 2
General comment
The article provides useful information about the effects of diet, muscle type and packaging on lipid oxidation and colour of Avileña Negra-Ibérica beef. Although, the experiment is in general appropriately designed and implemented, there are some points that should be corrected or clarified.
Major comments
Please use uniform description for vacuum packaging. VAC or FILM?
Actually there are two types of packaging, the colour measurement (Lightness, Chroma, Hue, redness, yellowness and discoloration) was made comparing vacuum packaging (VAC) and modified atmosphere packaging (MAP).
For visual color acceptability different steaks from the same joint from the objective color samples were used, in order to compare oxygen permeable film (FILM) and modified atmosphere (MAP) under display conditions
Why visual color acceptability was performed through photographs?
We decided to do the visual color acceptability through photographs because there were too much samples and we could not get enough people to make the experiment at the same time. We thought that the could be a possibility to carry out the analysis with all the samples through photographs, with each consumer doing the test at their own convenience. As previously explained to another reviewer, visual assessments with photographs have previously been reported with success (Cortez Pasetti et al., 2017. DOI: 10.1016/j.meatsci.2016.09.009; Cortez Pasetti et al., 2019. DOI: 10.1016/j.foodres.2019.03.036).
In Table 2, please let a space among columns or use two instead of three decimals
Ok, we have changed for better understanding.
Please explain what is “the percentage of positive assessments” in Material and Methods
The percentage of positive assessments, is the percentage of consumers who bought the steak of the picture.
Please merge paragraph L356-365 with that in L324-329 (repetition).
Done
Minor points:
L17: “serve as” instead of “be”
Done
L18: “…of the derived meat.”
Ok, it´s done
L19: “profitability” instead of “returns”
Ok
L24-25: Please delete “throughout display with either oxygen permeable film (FILM) or modified atmosphere (MAP)”
Ok
L26-27: “Display highly influenced lipid oxidation, since oxidation rates were lower by VAC than MAP packaging.”
Done
L28-29: What do you mean? Please rephrase
Ok, we have changed this sentence for better understanding
L37: “a variety of meat” instead of “differentiated”
Ok
L40: “The local breed of Avileña-Negra Ibérica is located in…”
Ok
L46: “…are also a feasible…”
Ok
L182: “considering” instead of “given”
Done
L199: Please rephrase
We have reworded this sentence
L214: “…was also affected by…”
Done, and we have rephrased that sentence
L217: “resulted in meat with lower red value” instead of “showed meat less red”
Done
L245: “…in VAC packaged meat. As it…”
We have changed it
L247: “…the inclusion of silage, with these values being higher in ST than LT muscle. Chroma value of MAP packaged meat was…”
We have changed this sentence for better understanding
L249: “…presented higher Chroma values than LT…”
Done
L250: “were not shown” instead of “disappeared”
Done
L265-266: “…at any packaging, with feeding being also significant under VAC…”
We have reworded this sentence
L276: “Figure 2.”
Ok, we have changed it.
L280: “affected” instead of “different”
Ok
L283: “Table 1” instead of “Figure 3”
We have changed it.
L332: VAC?
In that case it is FILM because these results are about visual colour acceptability not colour measurement.

Round 2
Reviewer 1 Report
Revisions are accpetable
Author Response
Thanks for your comments